# Dietary Patterns Related to Triglyceride and High-Density Lipoprotein Cholesterol and the Incidence of Type 2 Diabetes in Korean Men and Women

**DOI:** 10.3390/nu11010008

**Published:** 2018-12-20

**Authors:** Sihan Song, Jung Eun Lee

**Affiliations:** 1Department of Food and Nutrition, College of Human Ecology, Seoul National University, 1 Gwanak-ro, Gwanak-gu, Seoul 08826, Korea; songsihan@snu.ac.kr; 2Research Institute of Human Ecology, Seoul National University, 1 Gwanak-ro, Gwanak-gu, Seoul 08826, Korea

**Keywords:** dietary pattern, triglyceride, high-density lipoprotein cholesterol, type 2 diabetes

## Abstract

We aimed to examine whether dietary patterns that explain the variation of triglyceride (TG) to high-density lipoprotein cholesterol (HDL-C) ratio were associated with the incidence of type 2 diabetes in Korean adults. We included a total of 5097 adults without diabetes at baseline with a mean follow-up of 11.54 years. Usual diet was assessed by a validated food frequency questionnaire, and serum levels of TG and HDL-C were measured at baseline. We derived dietary pattern scores using 41 food groups as predictors and the TG/HDL-C ratio as a response variable in a stepwise linear regression. We calculated the odds ratio (OR) with the 95% confidence interval (CI) of type 2 diabetes according to pattern scores using multivariate logistic regression. A total of 1069 incident cases of type 2 diabetes were identified. A list of foods characterizing the dietary pattern differed by sex. Higher dietary pattern scores were associated with an increased risk of type 2 diabetes; ORs (95% CIs) comparing extreme quintiles were 1.53 (1.12–2.09; *p* for trend = 0.008) for men and 1.33 (0.95–1.86; *p* for trend = 0.011) for women. Our study suggests the evidence that dietary patterns associated with low levels of TG/HDL-C ratio may have the potential to reduce the burden of type 2 diabetes.

## 1. Introduction

The estimated global prevalence of diabetes mellitus has increased about 50% over the last decade, increased from 5.9% in 2007 to 8.8% in 2017 according to the International Diabetes Foundation [1,2] Over 90% of diabetes cases are of type 2 [2], and the rapid increase is largely attributed to changes in lifestyle factors, such as being overweight or obese, physical inactivity, unhealthy diet, and smoking [3,4]. To reduce the burden of diabetes and its complications, the early detection and treatment of diabetes and evidence-based guideline for diabetes prevention or management are important [5]. The World Health Organization (WHO) and the Food and Agriculture Organization (FAO) provide dietary recommendations for type 2 diabetes prevention, which include limiting saturated fat and obtaining an adequate fiber intake [6]. In addition, WHO recommends reducing the free sugar intake to prevent non-communicable diseases (NCDs), including type 2 diabetes [7].

Because nutrients and foods are consumed in combination, dietary pattern analysis can provide more practical evidence than single-nutrient analysis [8,9]. In addition, the consumption of dietary factors associated with the risk of type 2 diabetes may correlate with each other; therefore, dietary pattern analysis that reflects the complexity of diets may provide scientific insight and practical strategies for disease prevention. Approaches to assess the effects of dietary patterns on the risk of diabetes have been implemented in epidemiologic studies [10,11]. Several previous studies have found significant associations between the risk of type 2 diabetes and posteriori-derived dietary patterns that explain the variation in diabetes-related biomarkers, such as blood glucose, lipids, or inflammatory markers [12,13,14,15,16,17,18,19,20,21].

Insulin resistance, a phenomenon involving the resistance to insulin-stimulated glucose uptake, is involved in the pathogenesis of type 2 diabetes, hypertension, and coronary heart disease [22,23]. Indices derived from fasting glucose and insulin including homoeostasis model assessment-insulin resistance (HOMA-IR) and quantitative insulin-sensitivity check index (QUICKI) are widely used to quantify insulin resistance [24]. In addition, the triglyceride (TG) to high-density lipoprotein cholesterol (HDL-C) ratio has been suggested as a simple measure of insulin resistance [25,26,27,28]. Vega et al. [29] found that a high TG/HDL-C ratio was associated with an increased risk of type 2 diabetes and cardiovascular disease (CVD) mortality among men participating in the Cooper Center Longitudinal Study (CCLS). In the Korean population, the TG/HDL-C ratio was also significantly associated with insulin resistance [30,31], and the risk of type 2 diabetes [32].

The aim of our study was to identify posteriori-dietary patterns that explain variation in the TG/HDL-C ratio and to examine their association with the risk of type 2 diabetes in Korean adults. Given the sex-differences in TG and HDL-C levels [33] and dietary behavior [34], we identified the dietary patterns among men and women separately.

## 2. Materials and Methods

### 2.1. Study Population

The Ansan and Ansung study is part of the Korean Genome and Epidemiology Study (KoGES), which was designed to prospectively investigate the genetic and environmental influences on chronic disease in Korean adults [35]. The Ansan and Ansung study included 10,030 adults (4758 men and 5272 women) aged 40–69 years from the general population of urban (Ansan) and rural (Ansung) areas in 2001–2002. Clinical examination and interviewer-administered questionnaires were conducted at baseline and at biennial follow up. The follow-up rate was 62.8% at the 6th follow-up in 2013–2014 from baseline [35]. All participants provided informed consent. This study was approved by the Seoul National University Institutional Review Board (IRB No. E1811/001-009).

Of 10,030 participants at baseline, we excluded participants who had been diagnosed with or treated for diabetes (*n* = 683), cancer (*n* = 247), CVD (*n* = 408) or had missing data on relevant information (*n* = 7); those who had used insulin treatment (*n* = 93), an oral hypoglycemic drug (*n* = 360) or stroke medication (*n* = 21). We further excluded participants who did not have baseline fasting plasma glucose, 2-h plasma glucose after a 75 g oral glucose tolerance test (OGTT), or hemoglobin A1c (HbA1c) measurements (*n* = 67) as well as undiagnosed diabetes cases who met the American Diabetes Association (ADA) criteria for the diagnosis of diabetes (*n* = 692). We also excluded participants who did not have serum TG or HDL measurements (*n* = 1) or had outlier values for TG or HDL-C levels as determined by a box-plot method (greater than 3*interquartile range; *n* = 4).

KoGES provided food frequency questionnaires (FFQs) data after excluding individuals who: 1) did not answer any questions on the FFQs, 2) left more than 12 blanks for frequency questions, 3) did not answer any questions about rice intake, or 4) had extremely low (<100 kcal/day) or high (≥10,000 kcal/day) energy intake [36], resulting in exclusion of 255 participants. Furthermore, we excluded individuals who reported implausible energy intake (<500 or >3500 kcal/day for women and <800 or >4200 kcal/day for men; *n* = 230) or did not have data on alcohol consumption (*n* = 247). Among those participants who met the inclusion criteria at baseline (*n* = 7338), we excluded those who were not followed at the 5th (2011–2012) or 6th (2013–2014) follow up (*n* = 2228) as well as for those whose information on the ascertainment of type 2 diabetes during the follow-up (*n* = 754) was not available. As a result, a total of 5097 participants (2410 men and 2687 women) were included in the current study. Flow diagram of inclusion for study participants is presented in Appendix A.

### 2.2. Dietary Assessment

Usual dietary intake was assessed using an interviewer-administered semi-quantitative 103 item FFQ at baseline. The questionnaire was previously validated using 12-day dietary records among 124 participants of KoGES; Pearson’s correlation coefficients for energy-adjusted nutrient intakes ranged from 0.23 (vitamin A) to 0.64 (carbohydrate) [37]. Daily energy and nutrient intake were estimated based on the seventh edition of the Food Composition Tables of the Korean Nutrition Society [38]. Participants were asked to report the frequency and portion size of each food item during the previous year. Nine frequency categories ranging from “never or seldom” to “three times or more a day” and three portion sizes (small, medium, or large) were given as options. Alcoholic beverage consumption was assessed separately at baseline and at each biennial follow up. Alcohol drinking status was defined as nondrinker, past drinker, or current drinker. Current alcohol drinkers were asked to report the amount and frequency of alcohol beverage consumption in the previous month, and total alcohol consumption (g/day) was calculated based on the alcohol content of one standard drink.

### 2.3. Ascertainment of Type 2 Diabetes and Biomarker Assessment

A diagnostic test for diabetes and interviewer-administered questionnaires on diabetes diagnosis or treatment were repeated at biennial follow up. We defined incident type 2 diabetes cases as those who had: 1) a diagnosis of diabetes according to the ADA criteria including HbA1c ≥ 6.5% or fasting plasma glucose ≥ 126 mg/dl or 2-h plasma glucose ≥ 200 mg/dl after a 75 g OGTT [39]; or 2) a diagnosis of type 2 diabetes by physicians or treatment with insulin or oral hypoglycemic medication. Participants who developed type 2 diabetes after baseline were classified as diabetes cases and were otherwise categorized as non-cases.

Blood samples were collected after an overnight fast (at least 8 h) at baseline and at each biennial follow-up examination. Fasting plasma glucose, 1-h and 2-h plasma glucose after a 75 g OGTT were measured enzymatically using an automatic analyzer (ADIVA 1650; Siemens, USA). Whole blood HbA1c level was measured by high-performance liquid chromatography (BIO-RAD Variant II—Turbo; BIORAD, Japan). Serum TG and HDL-C were measured enzymatically using an automatic analyzer (ADIVA 1650; Siemens, USA). The TG/HDL-C ratio was calculated as the ratio of TG to HDL-C.

### 2.4. Covariate Assessment

Trained interviewers administered a questionnaire regarding sociodemographic factors, lifestyle factors, disease history or current treatment, medication history, family disease history, and reproductive factors. Smoking status was categorized into nonsmoker, past smoker, or current smoker. Pack-years of smoking were calculated using detailed information on smoking history among past or current smokers. When information regarding age at menopause was missing (*n* = 326), we considered the participant postmenopausal if they had been diagnosed after the age of 50, which was the median age of menopause in Korean postmenopausal women aged 40–69 years in 2001 [40]. Participants were asked to report: 1) the hours per day spent on four-intensity physical activity levels (sedentary, light, moderate, and vigorous activity) and 2) the frequency of leisure time spent on physical activity per week and the hours spent on each activity (aerobic, jogging, walking, swimming, tennis, golf, bowling, health club exercise, and mountain climbing). Physical activity was expressed as metabolic equivalents (METs) hours per week by multiplying the hours per week engaged in that activity by the activity’s corresponding MET value [41,42]. Anthropometric factors of body weight, height, waist circumference, hip circumference, and blood pressure were obtained by trained examiners. Body mass index (BMI) was calculated as weight (kg) divided by height-squared (m^2^).

### 2.5. Statistical Analysis

We derived dietary patterns that explained the variation in TG/HDL-C ratio using reduced rank regression (RRR) [43]. RRR is a posteriori method used to derive linear combinations of predictor variables (food groups) that explain as much as possible of the variation of response variables (disease-related markers) [43]. When using only one response variable, RRR is identical to multiple linear regression [12,44].

When we derived the dietary patterns of the participants at baseline, we included participants who had never been diagnosed with dyslipidemia and used hyperlipidemia drugs, and participants who had normal HbA1c and plasma glucose levels according to the ADA criteria to avoid the effect of these symptoms on their diet. As a result, 3630 participants (1716 men and 1914 women) were included in the study to derive dietary patterns.

We grouped 103 food items from FFQ and alcohol consumption into 41 groups (g/day) on the basis of similarities in food composition or nutrient content, and these were used as predictor variables to derive dietary patterns. As a response variable, the TG/HDL-C ratio was log-transformed for normality and adjusted for age using a residual method. We then derived dietary patterns that explained as much as possible of the variation of age-adjusted TG/HDL-C using a stepwise linear regression model in men and women separately. A significance level of 0.05 was used for entry and retention in the model. We calculated dietary pattern scores by summing the intakes of selected food groups that were weighted by the regression coefficients for all the study participants.

We used Spearman’s correlation coefficients to assess the correlations of dietary pattern and selected foods with the TG/HDL-C ratio. We divided the study participants according to the quintiles of dietary pattern scores and identified the intakes of selected food groups as well as demographic, lifestyle, and clinical characteristics. We used multivariate logistic regression models to calculate the odds ratio (OR) and 95% confidence interval (CI) of type 2 diabetes in each quintile of dietary pattern scores using the lowest quintile as the reference group. In multivariate models, we adjusted for age (continuous, years), living area (Ansan and Ansung), energy intake (continuous, kcal/day), menopausal status (pre and postmenopausal status for women), smoking status (0, >0 and <15, 15–<30, and 30≤ pack-years for men; ever and never smoking for women), alcohol consumption (0, >0 and <5, 5–<15, 15–<30, and 30≤ g/day for men; ever and never drinking for women), family history of diabetes (yes, no), chronic disease status at baseline (yes, no; diagnosis or use of medication for hypertension or hyperlipidemia), and physical activity (continuous, METs-hours/week). We further adjusted for BMI (kg/m^2^), which may be an intermediate factor in the causal pathway between dietary patterns related to TG/HDL-C ratio and the risk of type 2 diabetes. We calculated the p value for the trend across quintiles by assigning the median value of each quintile to corresponding participants and treating this value as a continuous variable in the model. We tested for effect modification by age, menopausal status, and BMI at baseline by performing stratified analyses and the likelihood ratio test (LRT) for each variable. We also conducted a sensitivity analysis by excluding incident cases of type 2 diabetes during 2 years of follow up when we examined the association between dietary pattern scores and the risk of type 2 diabetes. All statistical analyses were performed using SAS statistical software version 9.4 (SAS Institute Inc., Cary, NC). All hypothesis tests were evaluated using two-tailed tests of significance at *p* < 0.05.

## 3. Results

When we derived dietary patterns associated with TG/HDL-C ratio, the list of selected food items differed by sex (Table 1). A high dietary pattern score was characterized in men by higher intakes of noodles, fruits, fermented salted seafood and lower intakes of candy and chocolate, nuts, and pork, whereas that in women was characterized by higher intakes of organ and other meats and lower intakes of dairy products and nuts. The calculated dietary pattern scores were positively associated with TG/HDL-C ratio; the Spearman’s correlation coefficients were 0.15 for men and 0.13 for women.

The baseline characteristics of men and women are presented in Table 2 according to the quintiles of sex-specific dietary pattern associated with TG/HDL-C ratio. Men who had higher dietary pattern scores were more likely to live in a rural area, have a higher energy intake, and be a current smoker. Women who had higher dietary pattern scores were more likely to be older and postmenopausal, live in a rural area, and have a lower energy intake and physical activity. Participants who had higher dietary pattern scores were more likely to have a higher TG/HDL-C ratio, higher TG levels and lower HDL-C levels than those who had lower dietary pattern scores.

A total of 1069 (560 men and 509 women) cases of type 2 diabetes were identified over a mean follow up of 11.54 years. Multivariate adjusted OR (95% CIs) comparing extreme quintiles was 1.53 (1.12–2.09; *p* for trend = 0.008) among men and 1.33 (0.95–1.86; *p* for trend = 0.011) among women (Table 3). For women, OR (95% CI) comparing the 3rd versus the 1st quintile was 1.45 (1.06–1.99). After further adjustment for BMI, the association was slightly attenuated in both men and women. We examined whether the associations between dietary pattern scores and incident type 2 diabetes varied by age (≤median, >median), menopausal status, and BMI (<25, ≥25 kg/m^2^) at baseline regarding the risk of type 2 diabetes (Table 4). The associations appeared to be stronger among older or postmenopausal women than among younger or premenopausal women; the ORs (95% CI) comparing the extreme quintiles were 1.10 (0.65–1.86) for younger women and 1.61 (1.03–2.54) for older women. For BMI, the association appeared to be stronger among men with lower BMI than among those with higher BMI; the ORs (95% CI) comparing the extreme quintiles were 1.82 (1.19–2.80) for men with lower BMI and 1.17 (0.74–1.85) for men with higher BMI. However, there were no significant interactions by these factors. 

In the sensitivity analysis, we excluded incident cases that were identified during 2 years of follow-up (*n* = 193), and the estimates of the association between the dietary pattern scores and the incidence of type 2 diabetes were similar to those found in the main analysis (Appendix A).

## 4. Discussion

We derived dietary patterns that explain the variation in the diabetes-related biomarker, the TG/HDL-C ratio, in men and women separately. The dietary pattern was characterized by high intakes of noodles, fruits, fermented salted seafood and low intakes of candy and chocolate, nuts, and pork among men and by high intakes of organ and other meats and low intakes of dairy products and nuts among women. Dietary pattern scores were positively associated with the TG/HDL-C ratio. High dietary pattern score was associated with an increased risk of type 2 diabetes. The associations appeared to be stronger among older or postmenopausal women, albeit without statistical significance.

Insulin resistance is a major risk factor for the development of type 2 diabetes [22,23,45]. The consequences of insulin resistance and its compensatory hyperinsulinemia include glucose intolerance, dyslipidemia (increased TG and/or decreased HDL-C), high blood pressure, hyperuricemia, and increased plasminogen activator inhibitor (PAI)-1 activity [23]. Several studies have examined the association between risk of type 2 diabetes and dietary patterns that explain the variation of biomarkers that are linked to diabetes [12,13,14,15,16,17,18,19,20,21]. Biomarkers linked to diabetes that have been used to derive dietary patterns include inflammatory markers (e.g., PAI-1, tumor necrosis factor-α receptor 2, C-reactive protein (CRP), and Interleukin 6) [12,13,19,21], glucose (e.g., HbA1c, HOMA-IR, and fasting glucose) [12,14,16,20], lipid-related metabolites (e.g., TG, HDL-C, adiponectin, and leptin) [12,16,18,21], and uric acids [17]. Food groups that have been frequently reported to be associated with an increased risk of type 2 diabetes have been characterized by high intakes of sugar-sweetened beverages or soft drinks [12,13,14,16,17,21], processed meats [12,13,16,21], red meats [12,15,16,21], refined grains [13,16], white rice [21] or bread [14,19] and low intakes of wine [13,15,16], whole grains [16,21], and yellow [13,16,21] and green [16,21] vegetables. In our study, noodles positively contributed to the dietary pattern scores among men. Batis et al. [20] also identified a positive association of wheat noodles with dietary pattern scores that explained variations of HbA1c, HOMA-IR, and fasting glucose levels among Chinese men and women. Additionally, consistent with previous studies that identified a positive contribution of red meats to diabetes-related dietary pattern scores [12,15,16,21], we found a positive contribution of organ and other meats to dietary pattern scores among women.

The estimates of the association between dietary pattern scores and the risk of type 2 diabetes are comparable to those found in previous studies. Jannasch et al. [11] conducted meta-analyses of the association between posteriori-dietary patterns associated with diabetes-related biomarkers and risk of type 2 diabetes [12,13,14,16,46]; combined relative risks (95% CIs) were 0.51 (0.27–0.98) for dietary pattern related to HbA1c, HDL-C, CRP, and adiponectin [12,16,46], 2.53 (1.56–4.10) for dietary pattern related to inflammatory markers [13,16,46] and 1.39 (1.25–1.54) for dietary patterns associated with HOMA-IR [14,16,46].

Previous studies have identified the TG/HDL-C ratio as a clinically useful surrogate estimate of insulin resistance [25,26,27,28]. Methods for estimating TG and HDL-C concentrations are well standardized [26], and the ratio of the two values provides information on the atherogenic lipoprotein profile associated with the risk of CVD [47] as well as insulin resistance. A high TG/HDL-C ratio is associated with an increased risk of type 2 diabetes [29,32,48,49] and with CHD and/or CVD events [50,51] and mortality [29,52]. Tabung et al. [53] developed an empirical dietary index for insulin resistance using the TG/HDL-C ratio as a response variable and food groups as explanatory variables in a stepwise linear regression model in a female cohort, the Nurses’ Health Study (NHS). An empirical dietary index for insulin resistance was characterized by high intakes of low-calorie beverages, margarine, red meat, refined grains, processed meats, tomatoes, other vegetables, other fish, fruit juice, and creamy soups and low intakes of coffee, wine, liquor, beer, green leafy vegetables, high-fat dairy products, dark yellow vegetables, and nuts [53]. Consistent with a previous study that developed a dietary index for insulin resistance, our study also found that dietary pattern was positively associated with organ and other meats and negatively associated with dairy products and nuts among women. According to the meta-analysis of prospective cohort studies for individual food groups, a higher consumption of red meats was associated with an increased risk of type 2 diabetes [54], whereas higher consumption of dairy products [55,56] and nuts [57] were associated with a reduced risk of type 2 diabetes.

There were unexpected inverse associations of dietary pattern scores with pork and candy/chocolate among men. Even though we used an age-adjusted TG/HDL-C ratio, a difference in pork by age may exist. Low consumption of pork among older men may explain our finding; mean intake of pork among men aged 40s, 50s, and 60s was 47.19 g/day, 42.25 g/day, and 37.65 g/day, respectively. Further study is needed to examine dietary patterns associated with various biomarkers that have been linked to diabetes and their prediction of type 2 diabetes risk in different study populations.

This study has several limitations. We did not use person-years of follow-up to analyze the association because we did not have information on type 2 diabetes incidence for more than 20% of the original study participants. Therefore, we included participants whose fasting plasma glucose, 2-h plasma glucose after a 75 g OGTT, and HbA1c were available during the follow-up. This may limit the generalizability of our findings; however, we tried to remove the potential bias that could occur from loss to follow-up. We also cannot rule out the possibility of the presence of measurement error in the dietary assessment or residual or unknown confounding factors.

The strengths of our study include the measurement of biomarkers to detect the incidence of type 2 diabetes. Fasting plasma glucose, 2-h plasma glucose after a 75 g OGTT, and HbA1c were measured at baseline and at every biannual follow up. This is the first cohort study in Korea to examine the association between biomarker-related dietary pattern and type 2 diabetes incidence. We were able to examine a temporal relationship between dietary patterns and the risk of type 2 diabetes, given a cohort study design. Information on sociodemographic and lifestyle factors as well as medical status allowed us to adjust for potential confounders. Last, we additionally derived the dietary patterns of men and women separately. Providing sex-specific findings is important because of the sex difference in biological and behavioral characteristics and sex-specific results may provide appropriate evidence on modifiable factors that contributing to disease prevention in men and women [58]. We found that the components of dietary patterns differed by sex, and the results were consistent with previous studies that examined the association in a sex-specific way.

## 5. Conclusions

The present study derived posteriori-dietary patterns using the TG/HDL-C ratio as a biomarker that is linked to type 2 diabetes, and examined the associations between the dietary pattern scores and the risk of type 2 diabetes among Korean men and women. Among men, dietary pattern was characterized by higher intakes of noodles, fruits, fermented salted seafood and lower intakes of candy and chocolate, nuts, and pork; and the dietary pattern for women was characterized by higher intakes of organ and other meats and lower intakes of dairy products and nuts. Higher dietary pattern scores were associated with an increased risk of type 2 diabetes. Our study provides sex-specific evidence on dietary patterns associated with the risk of type 2 diabetes, which may partly be mediated by the TG/HDL-C ratio. Consideration of TG/HDL-C ratio related dietary patterns to reduce the burden of type 2 diabetes may be needed and our study warrants further replication.

## Figures and Tables

**Table 1 nutrients-11-00008-t001:** Spearman correlation coefficients between dietary pattern scores, selected foods, and the TG/HDL-C ratio, and the mean intake of selected food groups (g/day) according to the quintile of dietary pattern scores.

	Spearman’s Correlation with Dietary Pattern Scores		Quintile of TG/HDL-C Ratio-Related Dietary Pattern Scores		
	Dietary Pattern Scores	TG/HDL-C Ratio	Quintile1	Quintile3	Quintile5
Men (*n* = 2410)					
Diet pattern scores	1.00	0.15 ^a^			
Positive associations				Mean ± SD	
Noodles	0.37 ^a^	0.07 ^a^	63.24 ± 54.12	68.18 ± 47.97	152.57 ± 120.93
Fruits	0.33 ^a^	0.03	156.68 ± 150.35	154.63 ± 145.63	426.23 ± 382.61
Fermented salted seafood	0.23 ^a^	0.03	1.26 ± 2.66	0.88 ± 1.58	4.91 ± 8.12
Inverse associations					
Candy and chocolate	−0.30 ^a^	−0.06 ^a^	5.02 ± 8.62	0.68 ± 1.42	0.69 ± 1.90
Nuts	−0.29 ^a^	−0.06 ^a^	2.46 ± 4.40	0.39 ± 1.02	0.36 ± 1.08
Pork	−0.26 ^a^	−0.03	69.70 ± 55.90	32.44 ± 29.14	37.11 ± 34.88
Women (*n* = 2687)					
Diet pattern scores	1.00	0.13 ^a^			
Positive associations				Mean ± SD	
Organ and other meats	0.10 ^a^	−0.01	0.94 ± 3.11	0.97 ± 2.44	3.62 ± 16.72
Inverse associations					
Dairy products	−0.88 ^a^	−0.12 ^a^	292.40 ± 173.30	75.75 ± 40.48	9.51 ± 32.10
Nuts	−0.42 ^a^	−0.09 ^a^	2.50 ± 5.02	0.33 ± 0.57	0.04 ± 0.17

Abbreviations: TG, triglyceride; HDL-C, high-density lipoprotein cholesterol. ^a^ Spearman’s correlation coefficient was statistically significant (*p* < 0.05).

**Table 2 nutrients-11-00008-t002:** Baseline characteristics of the study participants according to the quintiles for TG/HDL-C ratio-related dietary pattern scores.

	Quintile of TG/HDL-C Ratio-Related Dietary Pattern Scores		
	Quintile1	Quintile3	Quintile5
Men (*n* = 2410)	482	482	482
Age (years), mean ± SD	50.38 ± 8.08	50.50 ± 7.99	51.10 ± 8.19
Residential area, *n* (%)			
Rural (Ansung)	148 (30.71)	209 (43.36)	260 (53.94)
Urban (Ansan)	334 (69.29)	273 (56.64)	222 (46.06)
Energy intake (kcal/day), mean ± SD	2095.97 ± 527.68	1802.31 ± 437.21	2289.32 ± 603.02
BMI (kg/m^2^), mean ± SD ^a^	24.02 ± 2.88	24.27 ± 2.77	24.37 ± 2.90
Physical activity (METs h/week), mean ± SD	9.65 ± 14.83	8.06 ± 12.29	8.76 ± 12.05
Smoking status, *n* (%) ^a^			
Non-smoker	116 (24.07)	97 (20.12)	99 (20.54)
Past smoker	175 (36.31)	165 (34.23)	130 (26.97)
Current smoker	191 (39.63)	220 (45.64)	253 (52.49)
Alcohol consumption status, *n* (%)			
Non-drinker	94 (19.50)	101 (20.95)	80 (16.60)
Past drinker	48 (9.96)	32 (6.64)	51 (10.58)
Current drinker	340 (70.54)	349 (72.41)	351 (72.82)
Family history of diabetes, *n* (%)			
No	434 (90.04)	433 (89.83)	437 (90.66)
Yes	48 (9.96)	49 (10.17)	45 (9.34)
TG/HDL-C ratio, mean ± SD	3.66 ± 2.79	4.14 ± 2.95	4.97 ± 4.06
TG (mg/dL), mean ± SD	150.52 ± 92.96	166.22 ± 92.86	191.56 ± 123.95
HDL-C (mg/dL), mean ± SD	44.88 ± 9.96	43.69 ± 9.27	42.73 ± 10.10
Women (*n* = 2687)	538	531	512
Age (years), mean ± SD	50.02 ± 8.06	50.24 ± 8.26	54.41 ± 8.97
Menopausal status, *n* (%) ^a^			
Pre-menopause	243 (52.83)	263 (55.96)	165 (35.71)
Post-menopause	217 (47.17)	207 (44.04)	297 (64.29)
Residential area, *n* (%)			
Rural (Ansung)	199 (36.99)	236 (44.44)	358 (69.92)
Urban (Ansan)	339 (63.01)	295 (55.56)	154 (30.08)
Energy intake (kcal/day), mean ± SD	2081.36 ± 513.71	1818.15 ± 480.73	1679.94 ± 517.76
BMI (kg/m^2^), mean ± SD ^a^	24.21 ± 2.97	24.72 ± 2.97	25.13 ± 3.23
Physical activity (METs h/week), mean ± SD	11.72 ± 16.47	9.46 ± 14.56	6.69 ± 10.16
Smoking status, *n* (%) ^a^			
Non-smoker	510 (96.05)	511 (96.78)	481 (95.06)
Past smoker	4 (0.75)	5 (0.95)	10 (1.98)
Current smoker	17 (3.20)	12 (2.27)	15 (2.96)
Alcohol consumption status, *n* (%)			
Non-drinker	368 (68.40)	378 (71.19)	371 (72.46)
Past drinker	10 (1.86)	14 (2.64)	18 (3.52)
Current drinker	160 (29.74)	139 (26.18)	123 (24.02)
Family history of diabetes, *n* (%)			
No	467 (86.80)	469 (88.32)	461 (90.04)
Yes	71 (13.20)	62 (11.68)	51 (9.96)
TG/HDL-C ratio, mean ± SD	3.08 ± 2.24	3.27 ± 2.64	3.74 ± 2.59
TG (mg/dL), mean ± SD	133.63 ± 74.45	136.19 ± 81.17	153.49 ± 82.92
HDL-C (mg/dL), mean ± SD	47.72 ± 10.1	45.59 ± 9.76	45.11 ± 9.95

Abbreviations: BMI, body mass index; METs, metabolic equivalents; TG, triglyceride; HDL-C, high-density lipoprotein cholesterol. ^a^ Some participants did not provide relevant information (among men, 1 missing for BMI; and among women, 326 missing for menopausal status and 27 missing for smoking status).

**Table 3 nutrients-11-00008-t003:** Odds ratios (ORs) and 95% confidence intervals (CIs) of incident type 2 diabetes according to the quintiles of TG/HDL-C ratio-related dietary pattern scores.

	Quintiles of TG/HDL-C Ratio-Related Dietary Pattern Scores					
	Quintile1	Quintile2	Quintile3	Quintile4	Quintile5	*p* for Trend
Men (*n* = 2410)						
Case/non-case	99/383	110/372	106/376	112/370	133/349	
Unadjusted model	Reference	1.14 (0.84–1.56)	1.09 (0.80–1.49)	1.17 (0.86–1.59)	1.47 (1.10–1.99)	0.010
Age-adjusted model	Reference	1.15 (0.85–1.57)	1.09 (0.80–1.48)	1.17 (0.86–1.60)	1.46 (1.08–1.97)	0.013
Multivariate adjusted model1 ^a^	Reference	1.17 (0.85–1.61)	1.12 (0.81–1.54)	1.19 (0.87–1.63)	1.53 (1.12–2.09)	0.008
Multivariate adjusted model2 ^b^	Reference	1.16 (0.84–1.59)	1.08 (0.78–1.49)	1.12 (0.82–1.53)	1.48 (1.09–2.03)	0.019
Women (*n* = 2687)						
Case/non-case	89/449	76/461	118/413	114/455	112/400	
Unadjusted model	Reference	0.83 (0.60–1.16)	1.44 (1.06–1.96)	1.26 (0.93–1.72)	1.41 (1.04–1.92)	0.002
Age-adjusted model	Reference	0.83 (0.59–1.15)	1.44 (1.06–1.95)	1.20 (0.88–1.64)	1.27 (0.93–1.74)	0.014
Multivariate adjusted model1 ^c^	Reference	0.83 (0.59–1.16)	1.45 (1.06–1.99)	1.23 (0.88–1.71)	1.33 (0.95–1.86)	0.011
Multivariate adjusted model2 ^b^	Reference	0.80 (0.57–1.13)	1.37 (1.00–1.89)	1.14 (0.81–1.59)	1.21 (0.86–1.70)	0.053

Abbreviations: TG, triglyceride; HDL-C, high-density lipoprotein cholesterol. ^a^ Adjusted for age (continuous, years), living area (Ansan and Ansung), energy intake (continuous, kcal/day), pack-years of smoking (0, >0 and <15, 15–<30, and 30≤ pack-years), alcohol consumption (0, >0 and <5, 5–<15, 15–<30, and 30≤ g/day), family history of diabetes (yes, no), hypertension or hyperlipidemia at baseline (yes, no), and physical activity (continuous, metabolic equivalents-hours/week). ^b^ Further adjusted for body mass index (continuous, kg/m^2^) in addition to the variables included in Model 1. ^c^ Adjusted for age (years), living area (Ansan and Ansung), energy intake (continuous, kcal/day), menopausal status (pre and postmenopausal status), smoking status (ever and never), alcohol consumption (ever and never), family history of diabetes (yes, no), hypertension or hyperlipidemia at baseline (yes, no), and physical activity (continuous, metabolic equivalents-hours/week).

**Table 4 nutrients-11-00008-t004:** Odds ratios (ORs) and 95% confidence intervals (CIs) of incident type 2 diabetes according to the quintiles of dietary pattern scores by age, menopausal status, and BMI at baseline.

	Quintiles of TG/HDL-C Ratio-Related Dietary Pattern Scores						
	Quintile1	Quintile2	Quintile3	Quintile4	Quintile5	*p* for Trend	*p* for Interaction
Men ^a^							
≤48 years, median (*n* = 1229)	Reference	1.26 (0.80–1.98)	1.02 (0.63–1.64)	1.12 (0.71–1.77)	1.65 (1.05–2.61)	0.052	0.788
>48 years (*n* = 1181)	Reference	1.09 (0.70–1.70)	1.21 (0.79–1.88)	1.26 (0.82–1.94)	1.46 (0.96–2.23)	0.063	
Women ^b^							
≤49 years, median (*n* = 1376)	Reference	0.73 (0.45–1.18)	1.20 (0.76–1.90)	1.02 (0.63–1.66)	1.10 (0.65–1.86)	0.407	0.319
>49 years (*n* = 1311)	Reference	0.90 (0.55–1.46)	1.75 (1.12–2.73)	1.49 (0.94–2.38)	1.61 (1.03–2.54)	0.004	
Menopausal-status at baseline ^c^							
Pre-menopause (*n* = 1143)	Reference	0.83 (0.49–1.41)	1.47 (0.90–2.41)	1.17 (0.68–2.01)	1.15 (0.64–2.07)	0.236	0.324
Post-menopause (*n* = 1218)	Reference	0.84 (0.50–1.41)	1.57 (0.97–2.55)	1.43 (0.87–2.33)	1.55 (0.96–2.50)	0.013	
Men ^a^							
<25 kg/m^2^ (*n* = 1441)	Reference	1.20 (0.78–1.85)	1.21 (0.78–1.89)	1.20 (0.76–1.89)	1.82 (1.19–2.80)	0.007	0.238
≥25 kg/m^2^ (*n* = 968)	Reference	1.12 (0.69–1.80)	0.92 (0.57–1.50)	0.98 (0.63–1.53)	1.17 (0.74–1.85)	0.593	
Women ^b^							
<25 kg/m^2^ (*n* = 1543)	Reference	0.63 (0.38–1.05)	1.53 (0.98–2.39)	1.04 (0.64–1.69)	1.17 (0.71–1.93)	0.138	0.625
≥25 kg/m^2^ (*n* = 1144)	Reference	1.00 (0.62–1.61)	1.31 (0.83–2.07)	1.31 (0.82–2.08)	1.29 (0.81–2.05)	0.133	

Abbreviations: TG, triglyceride; HDL-C, high-density lipoprotein cholesterol; BMI, body mass index. ^a^ Adjusted for age (continuous, years), living area (Ansan and Ansung), energy intake (continuous, kcal/day), pack-years of smoking (0, >0 and <15, 15–<30, and 30≤ pack-years), alcohol consumption (0, >0 and <5, 5–<15, 15–<30, and 30≤ g/day), family history of diabetes (yes, no), hypertension or hyperlipidemia at baseline (yes, no), and physical activity (continuous, metabolic equivalents-hours/week). ^b^ Adjusted for age (continuous, years), living area (Ansan and Ansung), energy intake (continuous, kcal/day), menopausal status (pre and postmenopausal status), smoking status (ever and never), alcohol consumption status (ever and never), family history of diabetes (yes, no), hypertension or hyperlipidemia at baseline (yes, no), and physical activity (continuous, metabolic equivalents-hours/week). ^c^ Adjusted for age (years), living area (Ansan and Ansung), energy intake (kcal/day), smoking status (ever and never), alcohol consumption status (ever and never), family history of diabetes (yes, no), hypertension or hyperlipidemia at baseline (yes, no), and physical activity (continuous, metabolic equivalents-hours/week).

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
