# Peer review of "Dietary Patterns Related to Triglyceride and High-Density Lipoprotein Cholesterol and the Incidence of Type 2 Diabetes in Korean Men and Women"

_nutrients, 2018, doi:10.3390/nu11010008_

Reviewer 1 Report

The paper has examined whether dietary patterns which can explain the variation in TG/HDL ratio are associated with the incidence of T2DM in adults. Please find attached, the PDF including my comments.

Line22-23: “Our study presents evidence that dietary patterns associated with the TG/HDL-C ratio may predict the risk of type 2 diabetes among Korean adults”.—what is the implication of this finding?

Line28: “2014”. — More recent statistics are available, based on the International diabetes federation. This needs to be updated.

Line29: “majority”.—How much?

Line110: “at least once after baseline”. —What does this mean?

Line124-129: Which questionnaire is this based on? Please provide info about its validity within T2DM population.

Line130: “blood pleasure”. —How was the BP measured?

Line148: “than”. —What does this mean?

Line190: “higher dietary pattern scores”. —In which dietary pattern?

Line301: a table providing frequency of consumption of foods would be useful.

Line301: conclusions—what is the implication of these findings? How these findings can help the field to move forward? What are your recommendations? Please elaborate.

Author Response

This is our reply to reviewer 1 comments.

We attached here the file.

Thank you.

Reviewer 2 Report

In this manuscript, dietary patterns related to triglyceride and high-density lipoprotein cholesterol and the incidence of type 2 diabetes in Korean men and women were investigated. The experimental design and results are good, which can support the conclusion. I have no further comments, just a minor question, please give the latest data of diabetes in line 27.

Author Response

This is our reply to reviewer 2 comments.

We attached here the file.

Thank you.

Reviewer 3 Report

In this manuscript Song and Lee reported an interesting finding that higher dietary pattern scores are associated with higher risk of developing type 2 diabetes in Korean adults, which is related to higher ratio of TG to HDL-C, and importantly that is sex-specific.

 Comments

1.     It might be better to draw a diagram or cluster that illustrates the screening and follow-up of the participants, so the readers can easily access it.

2.     I don’t quiet understand the “organ and other meats”, please explain.

3.     Change meters2 to m2 in line 132.

Author Response

This is our reply to reviewer 3 comments.

We attached here the file.

Thank you.
